# Assessing least-cost mitigation methods for environmental phosphorus loading of different pasture-based and housed dairy production systems in Great Britain

Brad P. Harrison[1], Martina Dorigo[2‡], Christopher K. Reynolds[1‡], Liam A. Sinclair[3‡], Ricard B. Tranter[4‡*], Partha P. Ray[1]

1 Department of Animal Sciences, School of Agriculture, Policy and Development, University of Reading, United Kingdom 2 Dairy, Agriculture and Horticulture Development Board, Stoneleigh Park, Kenilworth, Warwickshire, United Kingdom 3 Department of Agriculture and the Environment, Harper Adams University, Shropshire, United Kingdom 4 Department of Agri-Food Economics and Marketing, School of Agriculture, Policy and Development, University of Reading, United Kingdom

‡ These authors also contributed equally to this work.
☯ These authors contributed equally to this work.
* r.b.tranter@reading.ac.uk

## Abstract

Mitigating environmental phosphorus loading (EPL) from dairy farms reduces water pollution and improves the sustainability of production. Studies generally simulate EPL from dairy farms using a representative farm type from existing databases. However, housed and pasture-based dairy farming systems might contribute to eutrophication differently and have a varied feasibility of implementing mitigation. This study is the first that quantified EPL from dairy farms using data for FARMSCOPER collected from farmers and comparing EPL and identifying a least-cost suite of mitigation methods. Structural characteristics of 27 dairy farms in Great Britain (GB) were collected. Annual EPL from each farm was simulated in FARMSCOPER under three scenarios. Mean EPL of the production systems was compared to investigate any relationship between EPL and average 305 day adjusted milk yield of cows on each farm. A least-cost suite of mitigation methods was optimised for two model farms to represent either a housed or pasture-based system. Across both systems, 'current' implementation of mitigation methods was simulated to have reduced EPL from 0.63 to 0.56 kg P/ha (11%). The 'current' EPL positively correlated with milk production on a kg and kg/ha basis ($P \leq 0.001$ and $P = 0.033$, respectively). Farms operating a housed system had a mean 'current' EPL that was 59% greater than the pasture-based system though not significant ($P = 0.316$). This was partly due to a small sample size and because FARMSCOPER's estimates exclude variations in farm practices (i.e., feeding). EPL was reduced by ~50% and ~60% without incurring annual financial losses by implementing existing mitigation methods for pasture-based and housed systems, respectively. This study highlights the importance of mitigating EPL from GB dairy farming, especially considering the increasing number of higher yielding herds and housed production systems. Furthermore, emphasis should be on increasing implementation of system-specific mitigating methods; efforts to include more recent and specific farm data to improve the FARMSCOPER tool will benefit this.

**Data availability statement:** All relevant data are within the manuscript in S3 Table A listing of the data collected for analysis from the case–study dairy farms. From this is will be possible to replicate the results.

**Funding:** This research was funded by AHDB Dairy (for 41110062). PPR received the funding. The funders had no role in study design, data collection and analysis, decision to publish, or preparation of the manuscript.

**Competing interests:** The authors have declared that no competing interests exist.

## Introduction

Enrichment of phosphorus (P) in waterbodies accelerates eutrophication (degrading water quality and reducing aquatic biodiversity) which was estimated to incur a minimum annual loss of £229 million to the UK economy [1]. Since the amount of P loading to waterbodies from point sources (*i.e.,* sewage treatment works) has reduced over recent years, the diffuse sources of EPL (*i.e.,* agricultural land) are considered the most significant contributors to degrading water quality in Europe [2]. Therefore, the EPL from agriculture in the EU should be reduced in order to meet water quality objectives set out in the Water Framework Directive [3] by 2027 [4].

Mitigating EPL from GB dairy farming is increasingly important because more farms are using year-round housing in GB [5]. A year-round housed dairy farming system is modelled to pose a relatively higher eutrophic risk compared to a pasture-based system, primarily due to the import of a large amount of P in concentrate feeds [6,7]. The concept of increasing yields without causing environmental harm, and without acquiring more land, is considered as sustainable intensification. Pressures on agriculture in temperate regions to intensify sustainably are increasing due to the need for greater food production to satisfy a growing population whilst being constrained to a limited land area. Previous research reported that some innovative arable and mixed farms in GB have demonstrated sustainable intensification. However, achieving this with regard to P use in dairy farming was not observed [8]. On the contrary, the EPL from dairy farms in England was reported to positively correlate with production intensity [9]. However, these studies used data from before 2012 and, consequently, may not reflect current dairy farming systems. Therefore, there is a need to monitor progress towards achieving sustainable intensification in GB dairy farming by comparing the EPL from contemporary dairy farms with previous studies [8,9]. Any changes in EPL values could help indicate whether dairy farms are intensifying sustainably as regards P loading.

Nitrate Vulnerable Zones (NVZ) are designated in GB based on waterbodies containing more than 50 mg/l of nitrates. Farms within NVZs have mandatory restrictions on manure management and land application of Nitrogen [10], with the assumption that Nitrogen restrictions will also indirectly mitigate EPL. However, the effectiveness of NVZs in reducing EPL is uncertain because of the limited consideration for long-term accumulation of legacy P in the soil [11,12]. Policy-makers are increasingly interested in using voluntary approaches to influence positive environmental change [13]. For example, agri-environmental schemes such as the Countryside Stewardship Scheme in England, offer grants to farmers for the capital costs of implementing practices to improve the environment [14]. In particular, farmers in England and Ireland are reported to have a positive attitude towards changing practices associated with lower costs, such as those to reduce inputs [13,15]. Subsequently, the cost-effectiveness of individual mitigation methods of EPL relevant to GB agriculture has been explored using cost-curve analysis [16]. However, limited research has investigated the cost-effectiveness of suites of mitigation methods for GB dairy farming using a genetic algorithm approach (i.e., a search and optimisation technique inspired by natural evolution) [17]. Such an approach can overcome the short falling of a cost-curve approach, in regard to recognising a situation where it may be preferable to select one costly method over selecting a number of smaller methods with higher cost-effectiveness [18]. Consequently, there is need to investigate the cost-effectiveness of suites of methods to mitigate EPL from dairy farming using a genetic algorithm approach.

Previous studies using cost-curves recommended that further work is needed that investigates cost-effective mitigation options on a system-level [16]. Despite this, limited research has investigated suites of cost-effective methods to mitigate EPL from dairy farming on a system-level (i.e., pasture-based and housed). Dairy farming in GB operates a wide assortment of systems characterised by diverse calving approaches, varying amounts of concentrate

feeding and number of grazing days [19], and the feasibility of implementing practices that may differ between systems due to factors such as land availability and diet control (5). Therefore, there is a need to identify suites of least-cost methods to mitigate EPL from dairy farms on a system-level, to allow strategies to be developed to reduce EPL from modern, diverse GB dairy farming.

The 'FARM Scale Optimisation of Pollutant Emission Reductions' (FARMSCOPER) model was developed to simulate the diffused agricultural pollution from representative farm types. It is a Microsoft Excel-based decision support tool developed by the UK Government's Department of Environment, Food and Rural Affairs (DEFRA), that uses data on a farm's structure (*i.e.,* livestock and cropping) and physical characteristics (*i.e.,* soil type and rainfall) to simulate environmental loading of nutrients [18]. Additionally, FARMSCOPER can be used to optimize a least-cost suite of methods to mitigate pollutant loading by a targeted amount [20] using a dataset of mitigation methods and their impact on annual pollutant loading and their capital and operational costs [21]. This allows FARMSCOPER support decision making by policy-makers, whilst reducing the considerable costs of directly measuring EPL [18]. Therefore, it is important to ensure that FARMSCOPER produces accurate and reliable information on the EPL and least-cost methods to mitigate EPL in modern diverse dairy farming, if it is to continue to support the strategizing of mitigating EPL from dairy farms.

Previous studies have used FARMSCOPER to investigate the EPL from other farm types (i.e., dairy, arable and mixed), using existing datasets like the Agricultural Census [20], the Farmer Business Survey [9] and other published surveys [15] to gather data for use in FARMSCOPER. However, potential mismatches of existing datasets can require the transformation of data into an appropriate format, which involves some assumptions. Consequently, the use of existing datasets can provide less accurate and reliable inputs into FARMSCOPER compared to using a tailored approach (i.e., targeted surveys or focus groups) to directly collect appropriate data. A previous study used a tailored approach to specifically collect data readily appropriate for input into FARMSCOPER. However, such research collected data from only four dairy farms [8]. Therefore, there is need for information of EPL and least-cost mitigation methods for dairy farms, using FARMSCOPER input data collected directly from farmers using a tailored approach. Furthermore, there is a need to quantify the EPL and identify least-cost suites of methods to mitigate EPL from both pasture-based and housed dairy farming systems. The objectives of this study were to (1) quantify EPL from dairy farms using FARMSCOPER specific input data collected directly from dairy farmers using a tailored approach, (2) compare EPL data simulated from FARMSCOPER for housed and pasture-based dairy farming systems and (3) identify a least-cost suite of mitigation methods to reduce EPL from such dairy farming systems.

## Materials and methods

### Participating dairy farms

Dairy farm businesses from across GB were recruited through advertisements by various stakeholders (listed in the acknowledgements section below). Of the responding farm businesses, twenty-seven dairy farms with no other livestock enterprise were selected to ensure representation from a range of dairy farming systems [19]. Classification one farms adopted spring calving and grazed > 274 days a year with limited concentrate feed supplements. Classification two, three and four farms adopted block or all year calving with increasing use of concentrate feed supplementation as grazing days reduced. Classification five farms adopted all year round calving in a housed system with the greatest amount of concentrate use as a total mixed ration. For this study, classifications one (n = 4), two (n = 9) and three

(n = 7) were deemed pasture-based (a total of 20 farms) whereas classification four (n = 2) and five farms (n = 5) were deemed housed (a total of 7 farms). A similar number of dairy farms to a previous study (29 dairy farms) that collected data from large existing datasets [9] was achieved in the current study of 27 dairy farms. However, the number of participant dairy farms in the current study was considerably more than the four dairy farms used by the only other research study that similarly used a tailored approach to collect data specifically appropriate for FARMSCOPER directly from farmers [8]. Such a tailored data collection approach reduces the number of assumptions required and generates a more reliable data set [9,15,20].

## Data collection

Information on the farms' structure (i.e., livestock and cropping) and physical characteristics (i.e., soil type, rainfall) was collected by the lead author during visits from 25 November 2019 to 4 March 2020 using a pro-forma designed specifically to collect data appropriate for direct input into FARMSCOPER, thereby minimising the amount of assumptions required. This survey pro-forma had received permission for conduct through the lead author's Institution's (University of Reading, School of Agriculture, Policy and Development) Ethical Clearance Committee's procedure. All participants were provided with a Participant Information Sheet. This provided details about the project, what was required by the participant, how their data was to be stored, how they could withdraw their data, and by when. The Participant Information Sheet was supplied in advance of the visit of the researcher(s) for them to read it. When the researcher(s) visited the farm the participant was asked to sign the document with a written 'wet' signature before data collection began. The participants retained the Information Sheet whilst the researcher(s) retained the signed portion. Thus, the farmers gave informal written consent to the use of their data. Additionally, the dominant soil type for each farm's location was derived from Soilscapes [22], with soil types classified as freely draining considered as 'free draining' in FARMSCOPER. Slightly impermeable soils were considered as 'Drained for arable use', while impermeable soils were considered as 'Drained for grass and arable use'. Furthermore, rainfall data was determined for each farm's location using the same average precipitation data over 30 years that is used when calculating RB209 Nitrogen recommendations [23].

## Scenario analysis with FARMSCOPER

The FARMSCOPER tool is built on a suite of validated models that have been used in supporting UK policy-making [14]. Since the focus of this study was on P, the PSYCHIC model - Phosphorus and Sediment Yield Characterisation in Catchments [24,25], of FARMSCOPER was of particular importance. In the current study, FARMSCOPER was firstly used to simulate the annual EPL from each individual dairy farm by tailoring the customizable parameters in FARMSCOPER to match the farm's structure and physical characteristics. However, it is important to note that some variations in farm practices that are important in determining EPL (i.e., dietary P concentration) were fixed in FARMSCOPER. EPL for each farm was simulated under three scenarios: (1) 'baseline scenario' - when no mitigation methods are implemented; (2) 'current scenario' - when mitigation methods are implemented at the current rate [26] simulated by FARMSCOPER using national averages on the implementation of mitigation methods under existing schemes such as NVZs and the Countryside Stewardship Scheme; and (3) 'maximum scenario' - when all mitigation methods in the DEFRA user guide are implemented (i.e., regarding nutrient, livestock soil, delivery and pesticide management) [21].

The 'maximum scenario' expresses the maximum potential mitigation of EPL but excludes feasibility in terms of cost. Therefore, the optimisation feature within FARMSCOPER was

also used to identify the least-cost suite of methods to mitigate EPL by a minimum target of 5% of the baseline. FARMSCOPER optimises a selection of mitigation methods from within its list of mitigation methods which are characterised by their annual impact on pollutant loading and capital and operational costs. Optimisation occurs following the elitist NSGA-II genetic algorithm, which is an optimisation technique inspired by natural selection [27]. In FARMSCOPER, this algorithm is used to select the best solutions for a user-specified minimum target reduction of a specific pollutant at a minimum cost to the farmer. Essentially, this genetic algorithm approach operates on a population of artificial chromosomes, which represent a solution to a problem and has a fitness which measures how good a solution is to a particular problem. The genetic algorithm conducts fitness-based selection to produce a successor generation. The parents of each child solution are generated by tournament selection and solutions on the same Pareto front are given a higher probability of being selected to reproduce and survive into the next generation if neighbouring solutions are more distant [20]. This process continues for a specified number of generations, in which the most evolved solution is the optimal solution to the particular problem [17].

### Generation of model farms to represent a pasture-based and a housed dairy farming system

To utilise the optimisation feature of FARMSCOPER, previous studies generate a representative farm that is typical of one of the 17 representative farm types derived from the DEFRA 'Robust Farm Type' classification analysis [20]. However, the current study utilised the customizable parameters within FARMSCOPER to generate two model farms that closely represented either a pasture based or a housed dairy farming system, by using averages of the farm structure and physical characteristics from the participating dairy farms from each system (Table 1).

### Statistical analysis

The EPL simulated for each farm in FARMSCOPER was summarised using descriptive statistics in Minitab (Version 2019). Since the average herd size and UAA of participant farms were greater than their respective national averages, EPL was calculated on a total basis (kg) but also relative to UAA (kg/ha) and milk yield (kg/tonne milk). To compare EPL from previous studies, the EPL was also expressed as kg per unit of net energy (GJ) produced from milk production [8,9]. The energy content of milk was assumed to be 2.8 GJ of energy per 1000 litres of milk [8]. A linear regression analysis was used to investigate the relationship between the annual EPL and annual milk production for the farms on a total (kg and tonne, respectively) and a land use basis (kg/ha UAA and tonnes/ha UAA, respectively). The difference in mean EPL from farms operating a pasture-based vs housed system was investigated using ANOVA with mean separation by Tukey's test ($P \leq 0.05$ indicating significantly different means).

## Results

### Environmental phosphorus loading across all dairy farming systems under 'baseline', 'current' and 'maximum' scenarios

The mean annual EPL from all participant dairy farms (Fig 1), regardless of system, in the 'baseline scenario' was 114.5 kg (range = 13.8 to 583.6, S.E.M = 27.24) which equated to 0.63 kg P/ha UAA (range = 0.04 - 3.47, SEM = 0.130). Assuming that the implementation rate of on-farm mitigation methods used by FARMSCOPER in the 'current' scenario are representative of the participant dairy farms in the current study, farmers might have achieved a reduction in EPL of only ~ 11% from the 'baseline', equating to a 'current' EPL of 0.56 kg P/

**Table 1. Structural and physical characteristics of two model farms generated to closely represent a pasture-based and a housed dairy farming system.**

| Characteristic | | Pasture-based[a] | Housed[b] |
|---|---|---|---|
| **Livestock numbers** | | | |
| | Dairy cows | 254 | 219 |
| | Heifers | 71 | 85 |
| | Calves | 120 | 98 |
| **Land use** | | | |
| | Permanent pasture (ha) | 128 | 109 |
| | Rotational grazing (ha) | 51 | 0 |
| | Arable (ha) | 39 | 59 |
| **Soil Type** | | | |
| | | Free draining | Free draining |
| **Climate** | | | |
| | Rainfall (mm) | 900 - 1200 | 900 - 1200 |
| **Dirty water** | | | |
| | | Yard runoff and parlour washings sent to slurry store | Yard runoff and parlour washings sent to slurry store |
| **Grazing option** | | | |
| | | Access to watercourses while grazing | None |

[a]Generated using average (mean for continuous and mode for categorical) data of 20 surveyed farms.

[b]Generated using average (mean for continuous and mode for categorical) data of 7 surveyed farms.

ha UAA. However, the simulation under the 'maximum' scenario suggested the potential for a reduction in EPL of ~ 54% of the 'baseline', equating to a potential annual EPL of 0.29 kg P/ha through the implementation of all the existing mitigation methods in the DEFRA list [21]. (Below, in Table 3, the reader will see the seven mitigation methods that were selected as cost-effective methods of mitigating EFL).

The mean annual EPL under the 'baseline' scenario, per unit of milk produced and per unit of energy from milk produced were 0.057 kg//tonne of milk (range = 0.007 to 0.176, SEM = 0.0081) and 0.021 kg/GJ of milk per year (range = 0.003 to 0.065, SEM = 0.0030), respectively. The mean annual EPLs under the 'current' scenario, per unit of milk 0.0004 (range = 0.00003 to 0.002; SEM = 0.00085) kg/tonne of milk and per unit of energy from milk were 0.0001 (range = 0.00001 to 0.0008, SEM = 0.00031) kg/GJ of milk per year, respectively. The annual EPL from all participating dairy farms under both the 'baseline' and 'current' scenarios, positively correlated with total annual milk yield (tonnes) and annual milk yield relative to land use (tonnes/ha UAA) (Fig 2).

## Environmental phosphorus loading from pasture-based and housed dairy farming systems

A numerically lower ($P > 0.316$) mean EPL was predicted from the pasture-based system (Fig 3) compared to the housed system (Fig 4) under the 'baseline' (0.54 vs 0.84 kg P/ha, respectively), 'current' (0.49 vs 0.78 kg P/ha, respectively) and 'maximum' (0.25 vs 0.49 kg P/ha, respectively) scenarios. Consequently, this equated to a 56, 59 and 96% numerically higher mean EPL from farms using the housed compared to the pasture-based system under the 'baseline', 'current' and 'maximum' scenarios, respectively.

**Identifying a suite of least-cost methods to mitigate environmental phosphorus loading from a pasture-based and housed dairy farming system**

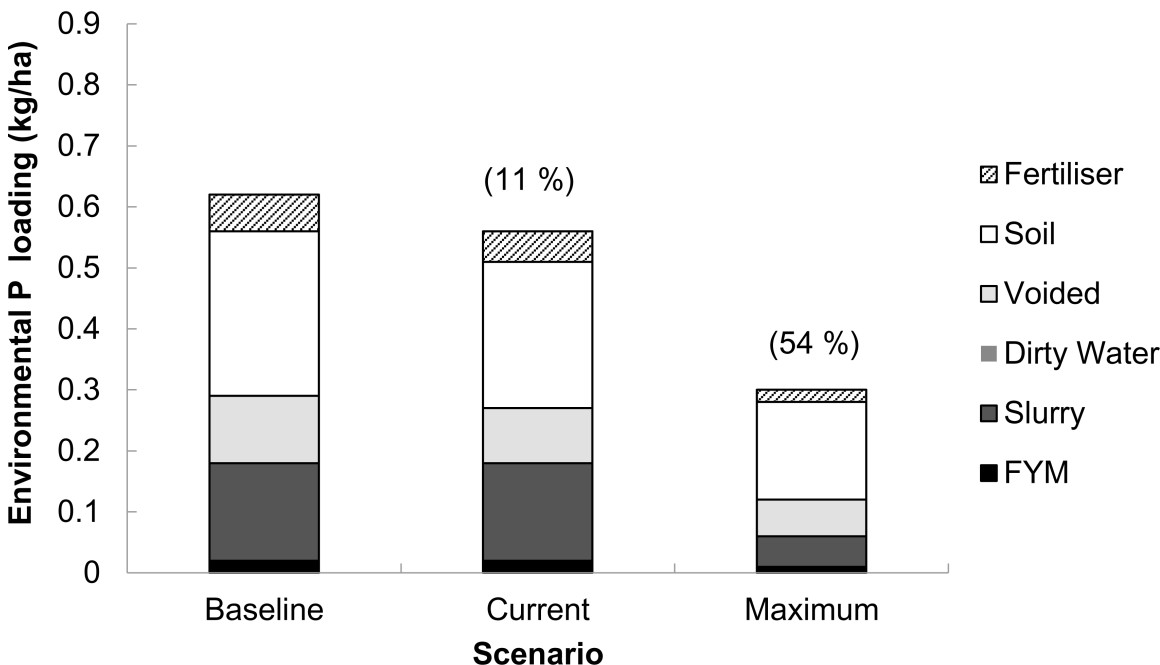

**Fig 1. Mean source apportionment of the annual environmental phosphorus (P) loading simulated in FARMSCOPER for 27 dairy farms in Great Britain across all systems.** 'Baseline' scenario - no mitigation methods implemented, 'Current' scenario –mitigation methods implemented at an estimated rate and 'Maximum' scenario - all mitigation methods in FARMSCOPER's dataset are implemented. Proportions (in parentheses) are the percentage reductions in environmental P loading from the baseline scenario.

**Table 2. Effect of the least-cost suites of mitigation methods that achieve minimum target phosphorus reductions for a pasture-based and housed dairy farming system.**

| Target reduction (%)[d] | Pasture–based[a] | | | Housed[b] | | |
|---|---|---|---|---|---|---|
| | Cost (£)[c] | Reduction achieved (%) | No. methods | Cost (£) | Reduction achieved (%) | No. methods |
| 5 | −45,578 | 25.6 | 26 | −74,176 | 15.4 | 14 |
| 10 | −45,190 | 17.8 | 23 | −64,788 | 34.6 | 24 |
| 15 | −46,394 | 21.3 | 21 | −60,097 | 32.7 | 25 |
| 20 | −48,093 | 21.4 | 25 | −69,430 | 28.3 | 22 |
| 25 | −44,393 | 26.2 | 23 | −68,926 | 37.5 | 26 |
| 30 | −41,538 | 31.5 | 26 | −67,854 | 34.7 | 21 |
| 35 | −31,941 | 35.1 | 31 | −59,119 | 39.6 | 31 |
| 40 | −20,551 | 42.9 | 28 | −53,872 | 40.8 | 29 |
| 45 | −11,288 | 45.2 | 34 | −55,114 | 45.2 | 29 |
| 50 | 2,790 | 50.0 | 34 | −42,783 | 50.2 | 28 |
| 55 | – | – | – | −17,643 | 55.6 | 31 |

[a]Generated using average (mean for continuous and mode for categorical) data of 20 surveyed farms.

[b]Generated using average (mean for continuous and mode for categorical) data of 7 surveyed farms.

[c]Total cost = captial cost + operational cost or saving.

[d]User specified minimum target of reduction (%) in EPL from the baseline EPL

The optimization feature of FARMSCOPER was first used to identify a range of cost-effective suites of methods to mitigate EPL from both the pasture-based and housed dairy farming system (Fig. 5). The pasture-based system could potentially reduce EPL by ~ 50%

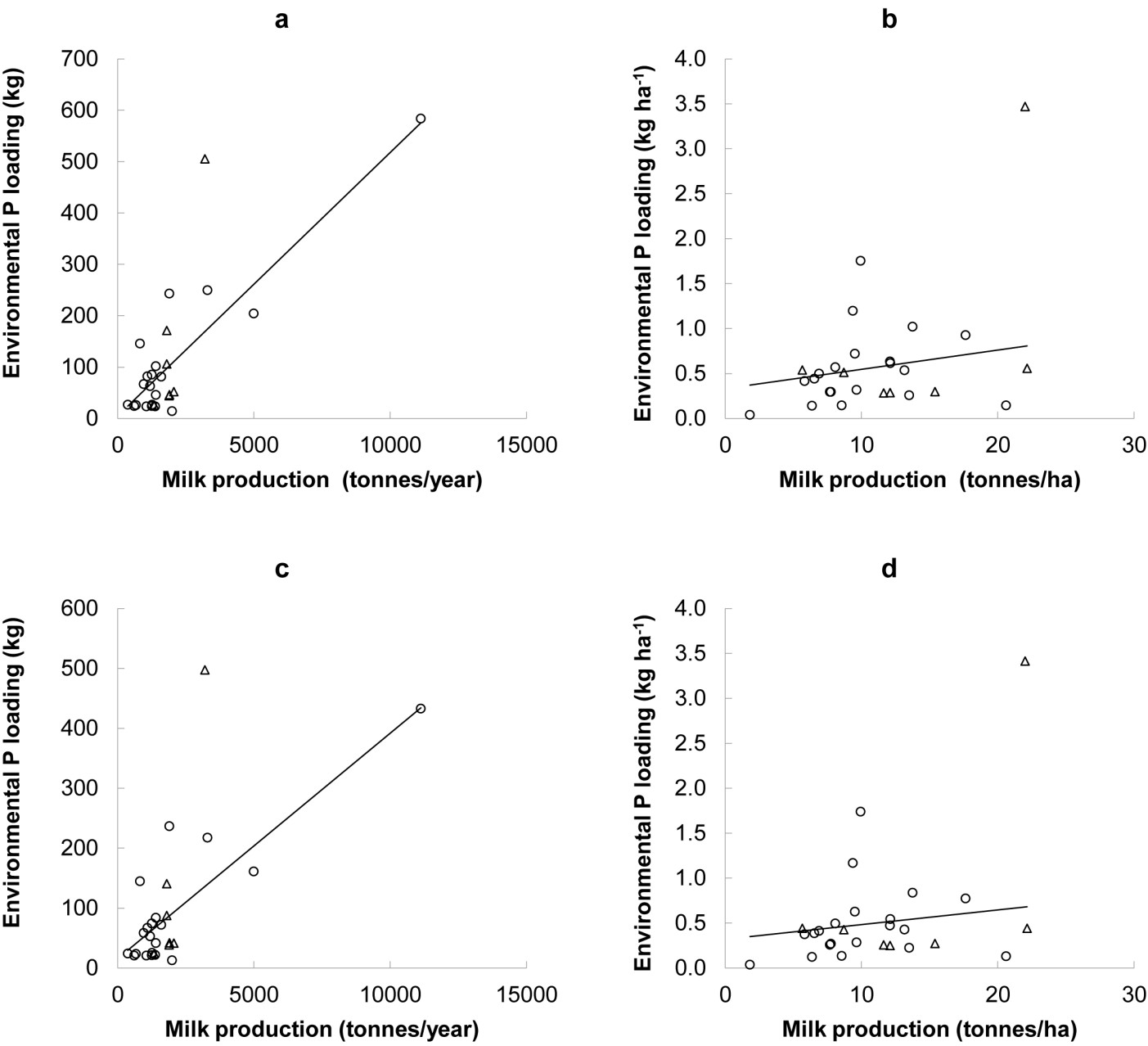

**Fig 2. Relationship between annual milk production and the annual environmental phosphorus (P) loading simulated using FARMSCOPER under the 'baseline' scenario ((a) total milk production/year (P ≤ 0.001; R² = 64.3%) and (b) milk production/year relative to land use basis (P = 0.026, R² = 18.1%)) and under the 'current' scenario (c) total milk production/year (P ≤ 0.001; R² = 49.4%) and (d) milk production/year relative to land use basis (P = 0.033, R² = 16.9%)).** 'Baseline' scenario - no mitigation methods implemented and 'Current' scenario –mitigation methods implemented at an estimated rate. Pasture-based dairy farming system (white circle; **n** = 20), housed dairy farming system (white triangle; **n** = 7).

of the 'baseline' without incurring annual financial losses, whereas the housed system could reduce EPL by ~ 60% without annual financial losses.

It was predicted that implementing the least-cost suite of 26 mitigation methods (S1 Table) to achieve the user-inputted minimum target of 5% reduction in EPL in the pasture-based system provided a potential annual saving of £45,578 and annual reduction of EPL

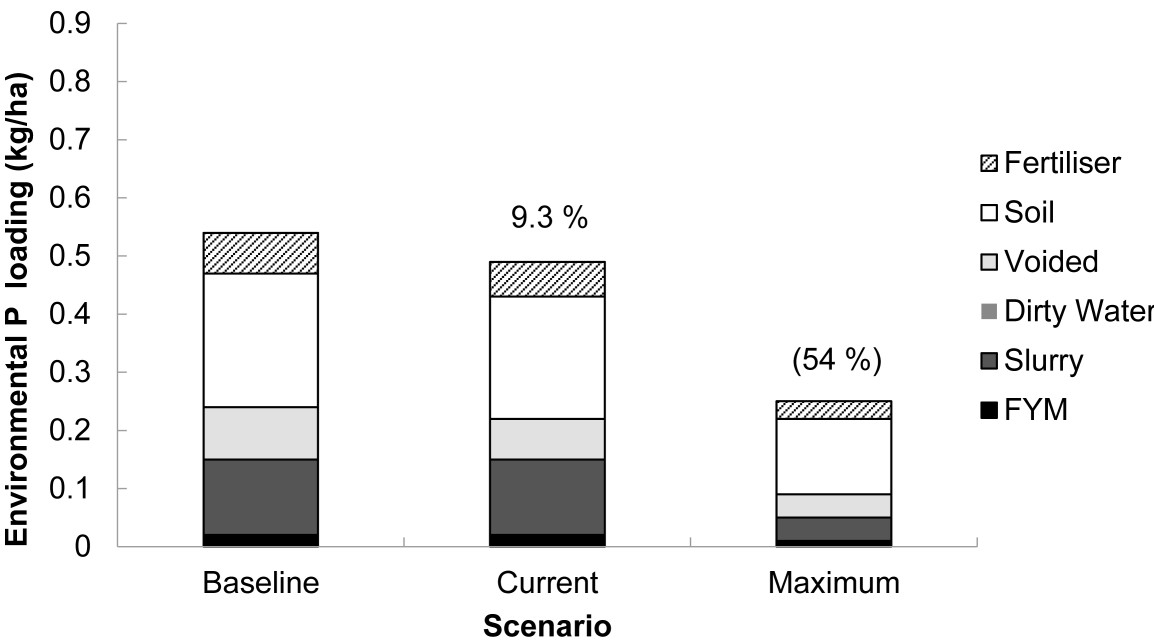

**Fig 3. Mean source apportionment of the annual environmental phosphorus (P) loading simulated in FARMSCOPER for farms operating a pasture-based system (n = 20).** 'Baseline' scenario - no mitigation methods implemented, 'Current' scenario - mitigation methods implemented at an estimated rate and 'Maximum' scenario - all mitigation methods in FARMSCOPER's dataset are implemented. Proportions (in parentheses) are the percentage reductions in environmental P loading from the baseline scenario.

by 25.6%. These savings were measured by computing the capital cost and operational cost or saving of producing the reduction in EPL. (Table 2). In contrast, a potential annual financial saving of £74,176 and a reduction of 15.4% in EPL when implementing the least-cost suite of 14 mitigation methods (S2 Table) to achieve the minimum target of at least a 5% reduction in EPL from baseline was indicated in the housed system. Across both dairy farming systems, the same seven mitigation methods were selected for every optimal suite of mitigation methods (Table 3).

## Discussion

### The representativeness of participating dairy farms

Across all systems, the farms in the current study had a larger mean herd size of 246 (78 to 920) lactating cows and utilised agricultural area (UAA) of 202 (64 to 920) ha, than the average 165 lactating cows and 154 ha UAA for typical GB dairy farms [28]. However, the mean annual milk yield of 7824 (4706 to 12091) kg/cow across all farming systems was similar to the 7889 kg/cow national average of GB dairy farms [29]. Therefore, since larger dairy farms (herd and land basis) are more aware of P pollution issues [30], consequently the current study may be reflective of dairy farmers that are relatively more interested in P management and thus may be reflective of a 'best case' situation.

Readers can consider the representation of the 27 study dairy farms by consulting the survey dataset presented as S3 Table below.

### Environmental phosphorus loading across all dairy farming systems under 'baseline', 'current' and 'maximum' scenarios

The broad range in EPL across all dairy farming systems under each scenario in the current study, suggested that the data collection approach sufficiently captured differences in farm structure

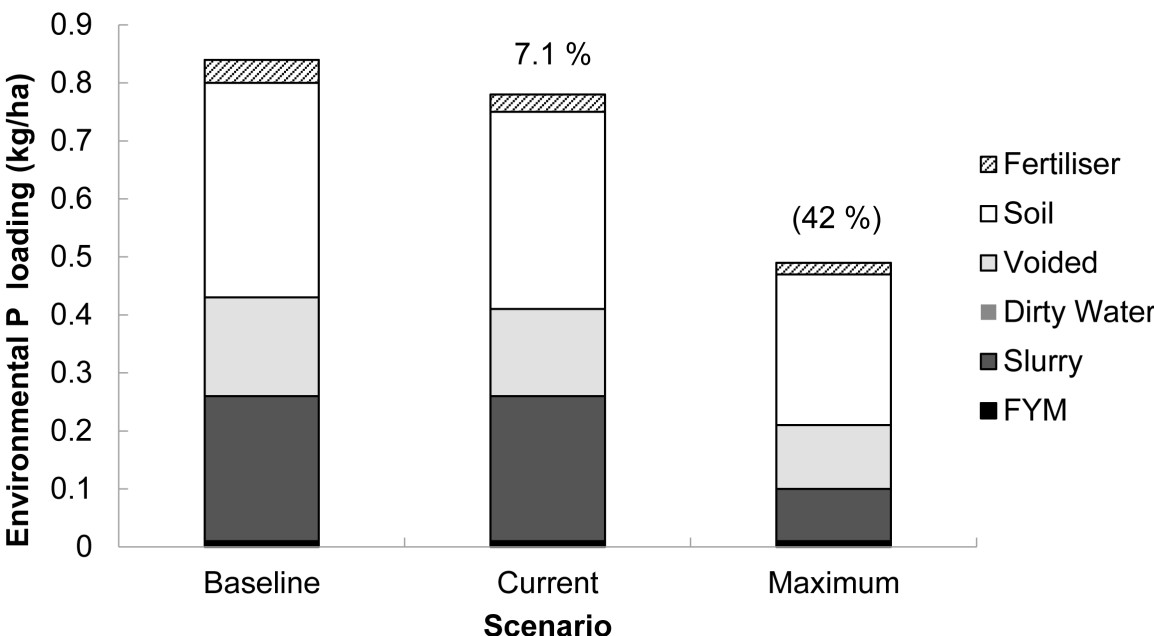

**Fig 4. Mean source apportionment of the annual environmental phosphorus (P) loading simulated in FARMSCOPER for farms operating a housed dairy farming system (n = 7) in Great Britain.** 'Baseline' scenario - no mitigation methods implemented, 'Current' scenario - mitigation methods implemented at an estimated rate and 'Maximum' scenario - all mitigation methods in FARMSCOPER's library are implemented. Proportions (in parentheses) are the percentage reductions in environmental P loading from the baseline scenario.

and physical characteristics that were important in determining EPL. However, the variation in the simulated EPL in this study that used a farm visit approach to collect specific data for model input could not be compared to prior studies that transformed data from existing datasets [9,20] because such studies did not provide information on the variation of simulated EPL from dairy farms. However, the mean annual EPL across all participating farms simulated for the 'baseline', 'current' and 'maximum' scenarios (0.63, 0.56 and 0.29 kg P/ha, respectively) in this study were all similar to the EPL simulated from dairy farms in the South of England using geo−referenced data, i.e., rainfall, soils and farm types specific for the Hampshire Avon test catchment [20] using the same scenarios (0.5, 0.44 and 0.19 kg P/ha). Conversely, EPL values in this study were lower than the mean 0.94 kg P/ha simulated from South−West England dairy farms using data adapted from the Farm Business Survey [9]. Therefore, findings in this study demonstrate the uncertainty associated with larger transformations of less relevant existing data sets into an appropriate format for inputting into models to simulate EPL. However, the implementation rate of various mitigation methods was not collected in this study and was assumed by FARMSCOEPR in the 'current' scenario by being simulated using older data on the implementation of mitigation methods under schemes and initiatives employed in the UK, such as NVZs and the Countryside Stewardship Scheme [26]. However, annual assessments of schemes such as Catchment Sensitive Farming report that there is an increase in the uptake of mitigation methods amongst farmers they advise [31].Consequently, the reliability of simulated EPL under the 'current' scenario could be improved by updating the average data used by FARMSCOPER, or by collecting additional information regarding the farm's actual implementation of mitigation methods [20].

The wide variation in EPL relative to milk production among the farms in the current study, supports the notion that there are opportunities for some dairy farmers to intensify

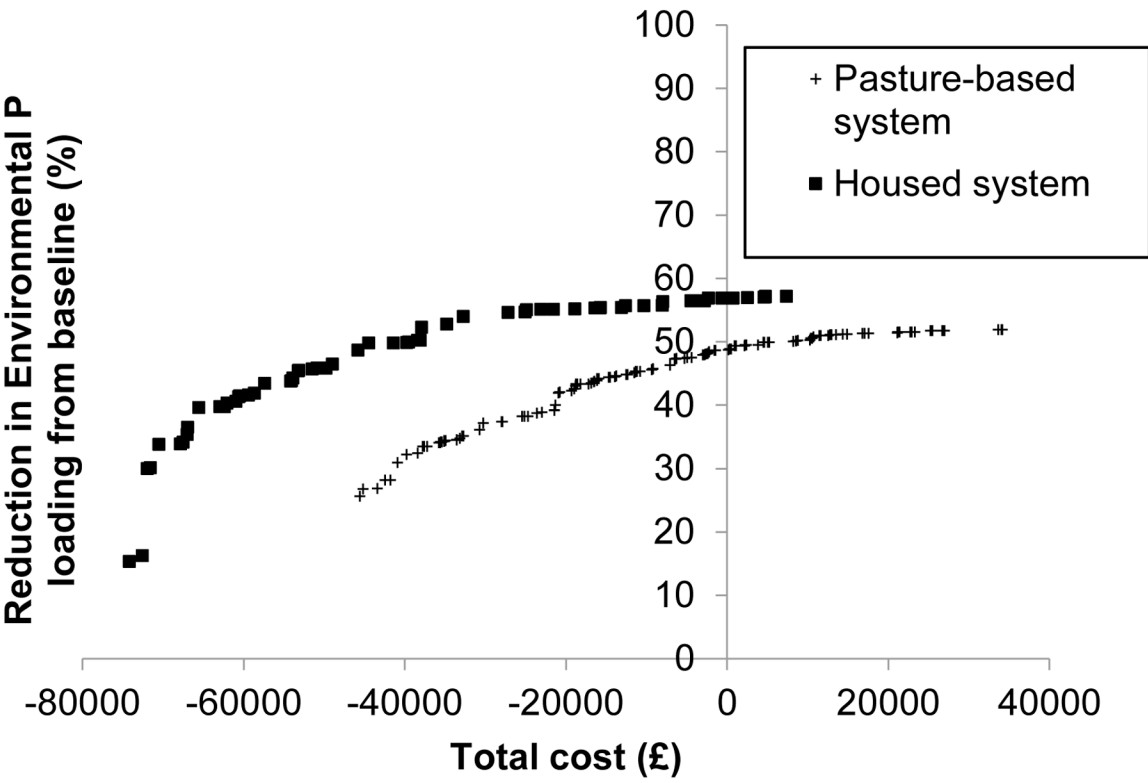

**Fig 5. Cost-effective suites of mitigation methods following optimisation on environmental phosphorus loading for a minimum target reduction of five percent, for two model farms generated to closely represent either a pasture-based[a] or housed[b] dairy farming system.** [a]Generated using average (mean for continuous and mode for categorical) data of 20 surveyed farms, [b]Generated using average (mean for continuous and mode for categorical) data of 7 surveyed farms.

**Table 3. Individual environmental and financial impact of the seven mitigation methods selected in all cost−effective suites of methods to mitigate environmental phosprus (P) loading from both a pasture−based and housed dairy farming system.**

| Mitigation method | Pasture−based[a] | | Housed[b] | |
|---|---|---|---|---|
| | P reduction (%) | Cost[c] (£) | P reduction (%) | Cost[c] (£) |
| Establish in−field grass buffer strips | 3.5 | 176 | 8.0 | 271 |
| Correctly−inflated low ground pressure tyres on machinery | 1.3 | −2,373 | 3.2 | −2438 |
| Management of arable field corners | 1.3 | 383 | 3.1 | 644 |
| Do not apply P fertilisers to high P index soils | 1.2 | − 730 | 2.6 | −630 |
| Make use of improved genetic resources[d] in livestock | 0.6 | −25,586 | 0.5 | −26,052 |
| Management of in−field ponds | 0.5 | 35 | 1.4 | 52 |
| Integrate fertiliser and manure nutrient supply | 0 | −13,928 | 0 | − 34,329 |

[a]Generated using average (mean for continuous and mode for categorical) data of 20 surveyed farms.

[b]Generated using average (mean for continuous and mode for categorical) data of 7 surveyed farms.

[c]Total cost = captial cost + operational cost or saving.

[d]Selection of genetic traits to allow increased productivity and fertility.

sustainably with regard to P [9], particularly when considering that farms producing similar amounts of milk had varying amounts of EPL. Therefore, farms with a higher EPL should aim towards operating with EPL values closer to the more environmentally sustainable dairy

farms of a similar milk production. The mean of 0.021 kg P/GJ milk produced per year of EPL across all farms under the 'baseline' scenario in this study, was slightly lower than the 0.03 kg P/GJ milk produced per year reported for South–West England dairy farms in 2012 using the same scenario [9]. The positive correlation between the annual energy of milk produced per ha and EPL per ha in this study ($R^2 = 0.17$) was weaker than the strength of the correlation ($R^2 = 0.53$) for dairy farms in South–West England in 2012 [9]. Therefore, the findings of this study indicate progress may have been made towards reducing phosphorus pollution from dairy farms from 2012 to 2019. However, this change over time may partly be attributed to differences in the samples of dairy farms used, or the transformation of data from an existing dataset into an appropriate format for input into FARMSCOPER by Lynch *et al.* [9]. Nevertheless, in this study the finding that EPL from dairy farms is positively correlated with the amount of milk produced, emphasises the importance of mitigating EPL from dairy farms, as average milk yield in GB continues to increase [5,32].

## Environmental phosphorus loading from pasture–based and housed dairy farming systems

Housed dairy farming systems are associated with increased imports of purchased concentrates, which usually contain 50% more P than grass herbage in GB [33]. Since the P concentration of manure is highly and positively correlated with dietary P intake in dairy cattle, a large amount of P–rich manure can be generated in a housed dairy farming system, which is then applied to adjacent arable and grass land usually in excess of crops' P requirement [6,34]. Applying P to land beyond the crops' requirement can result in soil P accumulation and subsequent environment P loading. Therefore, it has been suggested that housed dairy farming systems may be a significantly greater risk to EPL than pasture–based systems [6,7,35]. Conversely, although this study simulated a 59% greater mean annual EPL from the farms using a housed system compared to the pasture–based system under the 'current' scenario because of differences in livestock and land management and geographic (soil and rainfall) conditions, this numerical difference was not statistically significant ($P = 0.316$). The chances of finding significant differences in EPL between the housed and pasture–based dairy farming systems in this study were likely reduced because of the small sample size and because FARMSCOPER's estimates exclude variations in important farm practices (i.e., feeding).

FARMSCOPER uses a fixed grazing season of 117 days/year for dairy farms, which was raised as unrealistic by farm advisors in 2012 [36]. A shorter grazing season in housed systems results in a greater reliance on purchased concentrates [37]. Subsequently, the greater eutrophic risk associated with a housed system is largely attributed to their greater import of concentrate feed and subsequent higher manure P concentration [6]. FARMSCOPER is based on data from 2001 to 2007. However, the number of all–year housed systems amongst GB dairy farming has since increased [5]. Therefore, this study highlights the need for FARMSCOPER and other models of farm P flows to enable the manipulation of additional parameters in order for users to create a farming system that more closely matches their practice, if it is to continue to support farmers in making decisions about P management on their individual farms and policy decisions by simulating national and regional information that is reflective of modern diverse dairy farming systems.

## Least–cost phosphorus mitigation methods

In this study the optimization feature of FARMSCOPER suggested that there was considerable scope to reduce EPL by at least 50% in both farming systems without financial losses (capital expenditure being recovered through annual operational savings in some cases). Similarly,

previous studies that have investigated mitigation methods for various representative farm types using FARMSCOPER reported dairy farms to have the most pronounced net savings (capital – operational costs) when mitigating EPL compared to other farm types (13, 20). In this study, the same seven mitigation methods that were selected in every cost–effective suite of mitigation methods for both the pasture–based and housed system either targeted reducing nutrient input (i.e., integrating P concentration in manure and mineral fertiliser, make use of improved genetic resource and not applying mineral fertiliser P to high P index soils) to provide an operational saving, or were easy to implement (establish grass buffer strips, use correctly inflated low pressure tyres, manage arable field corners). Policy–makers are becoming increasingly interested in using voluntary approaches to influence positive environmental change [13], and farmers have been reported to have the most positive attitude towards changing practices that are associated with lower costs, i.e., practices that will reduce input use [13,15]. Therefore, the findings of this study suggest that more emphasis should be put on approaches to increase the implementation rate of existing mitigation methods.

The optimization of mitigation methods in FARMSCOPER is based solely on the environmental and financial impact given to each mitigation method in FARMSCOPER's library. Consequently, other important site–specific drivers of a mitigation method were not considered, such as the farmer's personal preference, technological innovation, agri–environmental scheme incentives and farm typology and practice [15,20]. Therefore, the feasibility of implementing the mitigation methods selected in the least–cost suite may vary with farm typology [15] and the financial saving for dairy farmers may also vary depending on factors such as agri–environmental incentives. In this study, differences in the mitigation methods selected in the least–cost suites occurred between the pasture–based and housed dairy farming systems. For example, increasing the use of maize silage in the housed system could provide potential annual operational savings, whereas soil management (i.e., loosening compacted soils) was important in reducing EPL in the pasture–based scenario but this was associated with an operational cost, consequently less annual financial savings occurred in the pasture–based system. Therefore, this study suggests that the approaches used to increase the implementation rate of existing methods to mitigate EPL in GB dairy farming would benefit from a system–specific approach.

## Opportunities to improve the accuracy of FARMSCOPER in predicting EPL and identifying a least–cost suite of methods to mitigate EPL

Since FARMSCOPER is a decision support tool which could be used to support policy–making, it is important to ensure that the results from FARMSCOPER simulation are accurate [14]. In this study, the greater potential financial saving associated with the least–cost suite of methods to mitigate EPL for the housed system compared to the pasture–based system was largely attributed to the method of integrating the P concentration of manure and fertiliser when planning land application rates. This was because of the greater production of manure in the housed system. Indeed, accurately crediting the P concentration of manure can provide financial savings by allowing more precise purchasing of mineral fertiliser P relative to manure P concentration [38]. However, integrating manure and fertiliser P may not be the most cost effective solution to reduce EPL for farmers handling P–rich manure in areas with a high soil P index, because farms may incur a cost to transport manure to more distant grass and arable land to avoid the risk of applying P in excess of the crops P requirement [38]. Therefore, lowering the concentration of P in manure by minimising the feeding of P in excess of the cows' requirement, which is a common practice on many GB dairy farms [39], is a recommended strategy [38].

In this study, FARMSCOPER only selected the method of 'reducing dietary P concentration' in ~ 25% of the cost−effective suites of methods to mitigate EPL, largely because it calculates the cost of reducing dietary P concentration by multiplying the number of dairy cows by a fixed factor of 0.02 and then multiplying this by an annual operating cost of £723. This calculation is derived from the assumption that more precise formulation of diets requires analytical data on forage P concentrations that is not readily available. Additionally, the calculation assumes that it is difficult to formulate low−cost, low−P diets because the P concentration in less expensive, protein−rich feed ingredients, which are commonly used in dairy cow diets, is considered high [21,40]. However, in many cases, P feeding could be minimised by simply eliminating or reducing the use of inorganic P supplements, which can provide financial savings [41] and minimise the water soluble fraction of manure P that is more prone to runoff [42]. In Northern Ireland, a field trial has observed that a reduction in the P concentration of diets fed to dairy cows from 5.4 to 3.0 g P/kg DM and applying the subsequently less P−rich manure from cows fed the lower dietary P concentration to land, significantly reduced the P concentration measured in overland flow. However, the observed large drop in P concentration of overland flow between simulated rainfall events suggested that increasing the time between manure application and the generation of over-land flow has a greater impact on P loss than does varying dietary P concentration [43].

Extending the grazing season was a selected method in the least−cost suite of methods to mitigate EPL for both the pasture−based and housed dairy farming system, largely because it provided an estimated saving in operational costs for farmers with regard to reduced cost of silage production and manure management [21]. However, extending the grazing season in an all−year housed system could reduce milk yield and have financial cost not necessarily consid-ered by FARMSCOPER. Conversely, FARMSCOPER also estimated that an extended grazing season would increase EPL because of increased soil poaching from grazing livestock [21]. EPL attributed to an extended grazing season may be lower than that simulated by FARM-SCOPER as the program does not consider the potential reduction in manure P concentration as a result of replacing a large amount of high P concentrate with grass−based feeds which typically contain 50% less P than concentrates in GB [37,44]. Furthermore, the method of extending the grazing season may not be feasible for a housed system where land for grazing is often limited. Therefore, this study highlights that further work into the annual environmen-tal and financial impact from the method of extending the grazing season could be important to improve the prediction accuracy of FARMSCOPER and subsequently FARMSCOPER's usefulness to farmers and policy−makers. Furthermore, this study supports that for decision support tools to be beneficial for policy−makers, they need to consider farm typologies to select the right measures at the farm−scale [15].

## Conclusions

The lower EPL simulated from dairy farms using appropriate data collected directly from farm-ers in this study compared to previous studies that simulated EPL from dairy farms using largely transformed data from existing datasets demonstrated the importance of considering the trade−off between a large sample size and uncertainty associated with larger data transformation. Fur-thermore, housed dairy farming systems in this study had a mean 'current' potential EPL ~ 60% numerically higher than those using a pasture−based system. Additionally, despite this study indicating progress towards improving the sustainability of dairy farming with regard to EPL, it also indicates that EPL from dairy farms will continue to be positively correlated with milk production on both a total yield and land basis. Therefore, this study emphasises the importance of ensuring effective mitigation of EPL as the prevalence of housed systems in GB dairy farming and milk yield have increased. This study also demonstrated that there is considerable scope to

reduce EPL by ~ 50% in a pasture−based and ~ 60% in a housed dairy farming system without incurring financial losses. These considerable reductions can be achieved by implementing existing mitigation methods. Therefore, the findings of this study suggest that more emphasis should be placed on approaches to increase the implementation rate of existing methods to mitigate EPL, such as increasing knowledge transfer between farmers, advisers and researchers. However, such approaches would benefit from a more system−specific approach based on farm typologies. Further consideration of the environmental and financial impacts from minimising excess P feeding and the increased customisability of parameters in FARMSCOPER and other P flow models are recommended to ensure that the results from simulations are reflective of modern GB dairy farming and, subsequently, provide accurate advice to policy−makers, farm advisers and farmers when developing strategies to mitigate EPL.

## Supporting information

**S1 Table. The 26 mitigation methods selected to achieve the minimum target of 5% reduction in environmental phosphorus (P) loading from a model farm generated to closely represent a pasture−based dairy farming system.** [a]Generated using average (mean for continuous and mode for categorical) data of 20 surveyed farms, [b]Total cost is the sum of capital and operational costs., [c]Total cost and reduction in environmental P loading may vary when evaluating mitigation methods individually compared to together
(DOCX)

**S2 Table. The 14 mitigation methods selected to achieve the minimum target of 5% reduction in environmental phosphorus (P) loading from a model farm generated to closely represent a housed dairy farming system.** [a]Generated using average (mean for continuous and mode for categorical) data of 7 surveyed farms, [b]Total cost is the sum of capital and operational costs, [c]Total cost and reduction in environmental P loading may vary when evaluating mitigation methods individually compared to together
(DOCX)

**S3 Table. A listing of the data collected for analysis from the case−study dairy farms.** [a] Classification one farms adopted spring calving and grazed > 274 days a year with limited concentrate feed supplements. Classification two, three and four farms adopted block or all year calving with increased use of concentrate feed supplementation as grazing days reduced. Classification five farms adopted all year round calving in a housed system with the greatest amount of concentrate use as a total mixed ration.
(XLSX)

## Acknowledgements

We thank all the participating dairy farmers and all consultants, organizations (Soil Association, Agricology, British Grassland Society, Royal Association of British Dairy Farmers, Maize Growers' Association) and staff at AHDB, SRUC, Bristol and Wessex Water, National Farmers' Union and the Environment Agency that helped in the recruitment of farmers for our study.

## Author contributions

**Conceptualization:** B.P. Harrison, C.K. Reynolds, P.P. Ray.

**Data curation:** B.P. Harrison.

**Formal analysis:** B.P. Harrison.

**Funding acquisition:** M. Dorigo, P.P. Ray.

**Investigation:** B.P.Harrison, R.B. Tranter, P.P.Ray.

**Methodology:** B.P. Harrison, M. Dorigo, C.K. Reynolds, L.A. Sinclair, R.B. Tranter, P.P. Ray.

**Project administration:** P.P. Ray.

**Supervision:** L.A. Sinclair, P.P. Ray.

**Writing – original draft:** B.P. Harrison.

**Writing – review & editing:** B.P. Harrison, M. Dorigo, C.K. Reynolds, L.A. Sinclair, R.B. Tranter, P.P. Ray.

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
