## [Decision Letter · Decision Letter 0]

25 Nov 2024

PONE-D-24-27234Assessing least-cost mitigation methods for environmental phosphorus loading of different pasture-based and housed dairy production systems in Great BritainPLOS ONE

Dear Dr. Tranter,

Thank you for submitting your manuscript to PLOS ONE. After careful consideration, we feel that it has merit but does not fully meet PLOS ONE’s publication criteria as it currently stands. Therefore, we invite you to submit a revised version of the manuscript that addresses the points raised during the review process.

We look forward to receiving your revised manuscript.

Kind regards,

Faham Khamesipour, Ph.D.

Academic Editor

PLOS ONE

Journal Requirements:

1. When submitting your revision, we need you to address these additional requirements. Please ensure that your manuscript meets PLOS ONE's style requirements, including those for file naming. The PLOS ONE style templates can be found at https://journals.plos.org/plosone/s/file?id=wjVg/PLOSOne_formatting_sample_main_body.pdf and 
https://journals.plos.org/plosone/s/file?id=ba62/PLOSOne_formatting_sample_title_authors_affiliations.pdf

For additional information about PLOS ONE ethical requirements for human subjects research, please refer to http://journals.plos.org/plosone/s/submission-guidelines#loc-human-subjects-research .

3. Thank you for stating the following financial disclosure: “This research was funded by AHDB Dairy (for 41110062).  PPR received the funding.”

4. We note that your Data Availability Statement is currently as follows: “All relevant data are within the manuscript and in Supporting Information files.”

Please confirm at this time whether or not your submission contains all raw data required to replicate the results of your study. Authors must share the “minimal data set” for their submission. PLOS defines the minimal data set to consist of the data required to replicate all study findings reported in the article, as well as related metadata and methods (https://journals.plos.org/plosone/s/data-availability#loc-minimal-data-set-definition). For example, authors should submit the following data: - The values behind the means, standard deviations and other measures reported; - The values used to build graphs; - The points extracted from images for analysis. Authors do not need to submit their entire data set if only a portion of the data was used in the reported study. If your submission does not contain these data, please either upload them as Supporting Information files or deposit them to a stable, public repository and provide us with the relevant URLs, DOIs, or accession numbers. For a list of recommended repositories, please see https://journals.plos.org/plosone/s/recommended-repositories. If there are ethical or legal restrictions on sharing a de-identified data set, please explain them in detail (e.g., data contain potentially sensitive information, data are owned by a third-party organization, etc.) and who has imposed them (e.g., an ethics committee). Please also provide contact information for a data access committee, ethics committee, or other institutional body to which data requests may be sent. If data are owned by a third party, please indicate how others may request data access.

Additional Editor Comments:

I would recommend a revision of the manuscript.

Reviewers' comments:

Reviewer's Responses to Questions

**Comments to the Author**

1. Is the manuscript technically sound, and do the data support the conclusions?

Reviewer #1: Yes

Reviewer #2: Yes

2. Has the statistical analysis been performed appropriately and rigorously? 

Reviewer #1: Yes

Reviewer #2: Yes

3. Have the authors made all data underlying the findings in their manuscript fully available?

Reviewer #1: Yes

Reviewer #2: Yes

4. Is the manuscript presented in an intelligible fashion and written in standard English?

Reviewer #1: Yes

Reviewer #2: Yes

5. Review Comments to the Author

Reviewer #1: Its very good manuscript for evaluate the P mitigation methods on different housing patterns in GB. Overall the manuscript is good and presented in intelligent fashion, however, some minor revisions have been identified as elaborated below.

1. Introduction is too long and may be summarized to a concise short one.

2. Reference 19 is missing in introduction section.

3. Some statements may be rephrased to be concise and meaningful (highlighted and pointed in the attached file).

4. Some grammatical mistakes are pointed out to be corrected.

5. Some statements may be replaced from MM section to Discussion.

6. In Fig.2 Why maximum scenario not included?

Overall it is a good effort.

Reviewer #2: This study is interesting and emphasizes the need to update the FARMSCOPER tool with more data sets. The report is well written, rigorous and easy to follow. I have only a very few suggested minor comments for the authors on the attached review document.

6. PLOS authors have the option to publish the peer review history of their article (what does this mean? ). If published, this will include your full peer review and any attached files.

**Do you want your identity to be public for this peer review?** For information about this choice, including consent withdrawal, please see our Privacy Policy .

Reviewer #1: **Yes: ** Muhammad Athar Abbas (DVM, PhD)

Reviewer #2: **Yes: ** J.G. Murnane

---

## [Author Response · Author response to Decision Letter 0]

24 Jan 2025

We have addressed all the reviewer and editor comments as requested and these are summarised in the Response to Reviewers document. The dataset with the information collected from the case-study farms in the analysis, is provided in new Supporting Table S3. As requested by the journal editor in paragraph 3, we have an amendment to make about Financial Disclosure - the role of the funder. The funder played a limited part in the design of the study. MD who, at the time of the study was employed by the funder, played a part in the preparation of the manuscript.

---

## [Decision Letter · Decision Letter 1]

11 Feb 2025

Assessing least-cost mitigation methods for environmental phosphorus loading of different pasture-based and housed dairy production systems in Great Britain

PONE-D-24-27234R1

Dear Dr. Tranter,

We’re pleased to inform you that your manuscript has been judged scientifically suitable for publication and will be formally accepted for publication once it meets all outstanding technical requirements.

Kind regards,

Faham Khamesipour, Ph.D.

Academic Editor

PLOS ONE

Additional Editor Comments (optional):

All comments have been addressed

Reviewers' comments:

Reviewer's Responses to Questions

**Comments to the Author**

1. If the authors have adequately addressed your comments raised in a previous round of review and you feel that this manuscript is now acceptable for publication, you may indicate that here to bypass the “Comments to the Author” section, enter your conflict of interest statement in the “Confidential to Editor” section, and submit your "Accept" recommendation.

Reviewer #1: All comments have been addressed

Reviewer #2: All comments have been addressed

2. Is the manuscript technically sound, and do the data support the conclusions?

Reviewer #1: Yes

Reviewer #2: Yes

3. Has the statistical analysis been performed appropriately and rigorously? 

Reviewer #1: Yes

Reviewer #2: Yes

4. Have the authors made all data underlying the findings in their manuscript fully available?

Reviewer #1: Yes

Reviewer #2: Yes

5. Is the manuscript presented in an intelligible fashion and written in standard English?

Reviewer #1: Yes

Reviewer #2: Yes

6. Review Comments to the Author

Reviewer #1: Dear Authors,

Thank you for submitting the revised version of your manuscript, PONE-D-24-27234R1. Your study on phosphorus mitigation in dairy farming is well-structured and makes a valuable contribution to sustainable agricultural practices. The use of FARMSCOPER modeling, cost-effectiveness analysis, and farm-specific data strengthens the scientific rigor of the paper.

While the manuscript is technically sound, a few minor revisions would enhance its clarity and impact. Below are specific suggestions for improvement.

To improve readability and conciseness, the following sentences should be rephrased:

• Line 29-31: This study is the first to quantify EPL from dairy farms using farmer-collected data for FARMSCOPER, comparing EPL levels, and identifying a cost-effective suite of mitigation methods.

• Line 66-68: Sustainable intensification refers to increasing yields while minimizing environmental impact without expanding land use.

• Line 612-617: The findings of this study indicate that greater emphasis should be placed on strategies to enhance the adoption of existing methods for mitigating EPL. This could be achieved through improved knowledge sharing among farmers, advisors, and researchers. Moreover, a more tailored approach, based on specific farm typologies, would be beneficial in optimizing such strategies.

Your manuscript is technically sound and contributes significantly to environmental phosphorus management in dairy farming. While these minor revisions are suggested to enhance clarity, structure, and discussion depth, it will be well-prepared for publication.

Best regards

Reviewer #2: (No Response)

7. PLOS authors have the option to publish the peer review history of their article (what does this mean? ). If published, this will include your full peer review and any attached files.

**Do you want your identity to be public for this peer review?** For information about this choice, including consent withdrawal, please see our Privacy Policy .

Reviewer #1: **Yes: ** Muhammad Athar Abbas (DVM, PhD)

Reviewer #2: **Yes: ** J.G. Murnane

---

## [Editor Report · Acceptance letter]

PONE-D-24-27234R1

PLOS ONE

Dear Dr. Tranter,

I'm pleased to inform you that your manuscript has been deemed suitable for publication in PLOS ONE. Congratulations! Your manuscript is now being handed over to our production team.

Kind regards,

on behalf of

Dr. Faham Khamesipour

Academic Editor

PLOS ONE